# Temperature and Thermal Energy of a Coronal Mass Ejection

Alessandro Bemporad 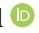

INAF-Turin Astrophysical Observatory, Via Osservatorio 20, 11025 Pino Torinese, TO, Italy;
alessandro.bemporad@inaf.it; Tel.: +39-011-8101-954

**Abstract:** Due to the scarcity of UV–EUV observations of coronal mass ejections (CMEs) far from the Sun (i.e., at heliocentric distances larger than 1.5 $R_{sun}$) our understanding of the thermodynamic evolution of these solar phenomena is still very limited. This work focuses on the analysis of a slow CME observed at the same time and in the same coronal locations in visible light (VL) by the MLSO Mark IV polarimeter and in the UV Lyman-$\alpha$ by the SOHO UVCS spectrometer. The eruption was observed at two different heliocentric distances (1.6 and 1.9 $R_{sun}$), making this work a test case for possible future multi-slit observations of solar eruptions. The analysis of combined VL and UV data allows the determination of 2D maps of the plasma electron density and also the plasma electron temperature, thus allowing the quantification of the distribution of the thermal energy density. The results show that the higher temperatures in the CME front are due to simple adiabatic compression of pre-CME plasma, while the CME core has a higher temperature with respect to the surrounding CME void and front. Despite the expected adiabatic cooling, the CME core temperatures increased between 1.6 and 1.9 $R_{sun}$ from 2.4 MK up to 3.2 MK, thus indicating the presence of plasma heating processes occurring during the CME expansion. The 2D distribution of thermal energy also shows a low level of symmetry with respect to the CME propagation axis, possibly related with the CME interaction with nearby coronal structures. This work demonstrates the potential of UV and VL data combination and also of possible future multi-slit spectroscopic observations of CMEs.

**Keywords:** sun: atmosphere; sun: corona; sun: UV radiation; sun: coronal mass ejections (CMEs)

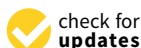



## 1. Introduction

For decades the origin and the early evolution of plasma embedded in coronal mass ejections (CMEs) have been studied via remote sensing observations [1,2]. The CME early evolutions are typically studied with space- and ground-based coronagraphs such as the instruments (Mark IV, K-Cor, CoMP) at the Mauna Loa Solar Observatory (MLSO) [3], the LASCO coronagraphs on board SOHO [4,5], and the SECCHI COR1 and COR2 instruments on board STEREO [6,7]. All these instruments measure the visible light (VL) emission, which is due to Thomson scattering of photospheric radiation by coronal electrons [8], and hence provides estimates of the local plasma column density and number density [9] (once the emission from the F-corona is removed [10]), as well as the projected expansion velocities by tracking specific features and applying image filtering analysis [11,12], or even the un-projected velocities by combining observations from different points of view [13,14].

Nevertheless, these instruments cannot provide measurements of many other CME plasma physical parameters such as plasma temperatures and elemental composition; this information can be extracted only with UV–EUV and X-ray observations [15] acquired from space. For this reason, many eruptions have been studied with UV–EUV imagers such as the EIT telescope on board SOHO [16], the STEREO EUVI instruments [6], the AIA telescopes on board SDO [17], and the SWAP instrument on board PROBA2 [18]. Particularly as a result of the development of differential emission measure (DEM) techniques applied to EUV images [19], together with the development of atomic spectral line databases [20], it is now possible to analyze these images to infer the 2D distribution of plasma temperatures inside CMEs [21,22]. In most cases, these images were also complemented with data acquired by

UV–EUV spectrometers such as EIS on board Hinode [23], and the SUMER [24], CDS [25], and UVCS [26] spectrometers on board SOHO.

Among all the different UV–EUV spectrometers that observe the solar corona, there is only one instrument that is able to study the early expansion phases of CME in the intermediate corona (from 1.5 up to $\sim$6–10 $R_{sun}$): UVCS on board SOHO. This is due both to the large projected extension of the spectrometer slit field of view (FOV), covering 40 arcmin, and to the unique capability of this instrument to observe at very different heliocentric distances from 1.5 up to 10 $R_{sun}$. This is the very important coronal region where CMEs develop from the initial acceleration/impulsive phase (observed by disk imagers and spectrometers) to the subsequent propagation phase (observed by coronagraphs and heliospheric imagers), thus starting their final transition to becoming interplanetary CMEs. The most complete information on the early evolution of CME plasma physical parameters (densities and also temperatures and elemental abundances) has been derived from the intermediate corona via the UVCS observations, leading to unexpected discoveries (see the review by [27]).

In particular, the analysis of many different events observed by UVCS demonstrated that an additional heating source for the CME plasma must be considered during the early expansion in order to reproduce the observed UV–EUV emissions, i.e., a source providing a total thermal energy comparable with those reported in [28,29] or in most cases several times larger than those in [30,31], the total kinetic and potential energies dragged by the eruptions. The origin of this additional source of energy is still unknown, but theoretical considerations [32] suggest that it could be due to the conversion of magnetic energy into plasma heating via anomalous resistivity related to magnetic reconnection occurring in the expanding flux rope, as predicted by [33].

These observations are very interesting for many reasons. It is usually very difficult to estimate in the intermediate corona this significant fraction of the CME plasma energy to be added to the kinetic and potential energies usually estimated with VL coronagraphs. Analyses of EUV images and X-ray observations in the inner corona are employed by many authors to relate the flare energy release with the initial CME plasma heating [34–37], and these results seem not to be affected by the usual assumption of ionization equilibrium [38]. Nevertheless, very little is known about the subsequent CME plasma thermodynamic evolution above $\sim$1.5 $R_{sun}$.

Most of the recent thermodynamic models of CME expansion derive the thermal energy evolution in the hypothesis of adiabatic expansion by neglecting possible internal sources for plasma heating [39], but empirical estimates of the polytropic index $\gamma$ in ICMEs point to values of the order of 1.15$\sim$1.33, implying considerable local plasma heating. In fact, the need for CME plasma heating occurring until the ionization states are frozen in has been found from in situ data [40], and additional plasma heating sources are required to explain the lower-than-expected proton temperature decrease in ICMEs [41,42].

All the above considerations motivated the work presented here, which is based on a further analysis of the UVCS data already described by [43] (hereafter Paper I), analyzed using the revised technique recently described by [44] (hereafter Paper II) conceived to measure the solar wind speed and instead applied here for the first time to measure the temperatures inside a CME. The uniqueness of these observations is related to the fact that it is possible not only to estimate the thermal energy of a CME during its early propagation phase but also to derive for the first time the temperature evolution inside the CME between two different altitudes. The results presented here show, interestingly, that during the CME expansion the core temperatures are increasing. After a description of the data (Section 2), the results are presented (Secion 3) and finally discussed in the light of current and future space missions (Secion 4).

## 2. Data Description and Analysis

As in previous work by, for instance, [45], this study combines contemporaneous and cospatial observations of the same eruption in two different wavebands: the UV and

VL. These types of combined observations are very important, because different plasma parameters can be derived independently. In particular, the plasma electron density $n_e$ can be derived directly from the VL. Then, once the density is determined, this can be combined with the UV–EUV observations to measure the plasma electron temperature $T_e$. Nevertheless, the use of images acquired by EUV imagers usually limits these kinds of analyses to heliocentric distances not larger than $\sim$1.5 $R_{sun}$.

The data analyzed here were acquired by the UVCS instrument with the slit center located at two different heliocentric distances: 1.6 $R_{sun}$ and 1.9 $R_{sun}$. The uniqueness of these observations lies in the fact that the exposure sequence was acquired by moving the spectrometer slit alternately to the two altitudes. In particular, on 31 January 2000 the UVCS observations started at 17:05 UT, ending on 1 February at 02:00 UT. The projected FOV of the UVCS slit was centered at a northern latitude of 60° in the north-east quadrant (see Figure 1). The instrument took alternately 10 exposures at 1.6 $R_{sun}$ and three exposures at 1.9 $R_{sun}$, with an exposure time of 120 s (see Paper I for more details). The data analyzed here were acquired with the so-called "O VI channel" (optimized for observations in the spectral range around the O VI 1031.90 Å/1037.63 Å doublet), covering the spectral interval between 984 Å and 1080 Å. The spatial resolution (considering the binning) was 42 arcsec/pixel.

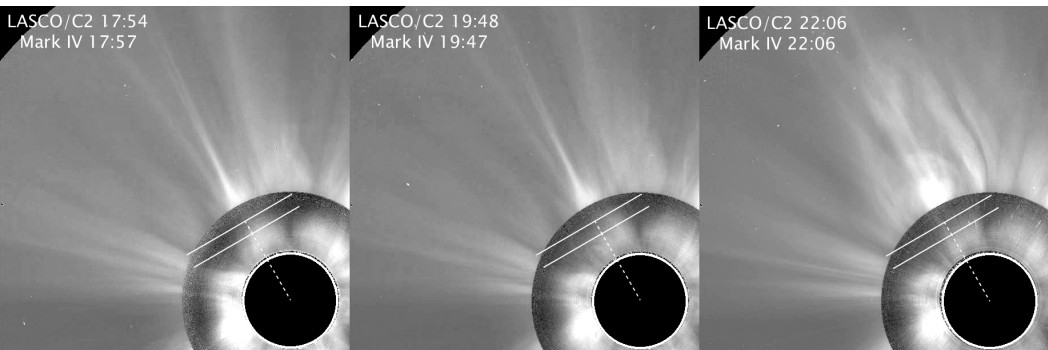

**Figure 1.** The 31 January 2000 CME as observed by the MLSO Mark IV K-coronameter (inner coronal regions) and by the SOHO/LASCO-C2 coronagraph (outer coronal regions). The white straight lines mark the projected locations of the UVCS slit field of views, centered at 1.6 and 1.9 $R_{sun}$ at a latitude of 60° north.

On the same day, the MLSO Mark IV K-coronameter measured the VL polarized brightness (pB) with 91 exposures covering the time interval from 17:30 UT to 22:07 UT; however, the first three exposures were corrupted and were not used in the analysis. The Mark IV K-coronameter acquired images of the low corona (700–1080 nm) from $\sim$1.12 to $\sim$2.79 $R_{sun}$ every $\sim$3 min with an angular resolution of 0.5°, providing a 960 × 960-pixel image of the corona. Given the uncertainties of the Mark IV instrument [3], pB values are only reliable below $\sim$2 $R_{sun}$.

As shown in Figure 1, starting from $\sim$18 UT on 31 January 2000, a slow CME started to erupt in the north-east quadrant. The speed of the CME front (as determined by a second-order fit of the height versus the time curve obtained from LASCO and Mark IV data—see Paper I) increased from $\sim$30 km s$^{-1}$ at 1.6 $R_{sun}$ (18:30 UT) up to $\sim$160 km s$^{-1}$ at 2.6 $R_{sun}$ (20:00 UT), while the speed of the core showed a large spread of values between 70 and 100 km s$^{-1}$ but no significant acceleration. This event can be thus classified as a slow CME; these speed values are very important also in the analysis of UVCS data, as briefly explained below. Due to these low speed values, the alternate acquisition of UVCS observations at 1.6 $R_{sun}$ and 1.9 $R_{sun}$ allowed the observation of the same event at two different altitudes. Hence, these observations can be considered as an example of a "multi-slit-like" observation of a CME, similar to that originally proposed for the METIS coronagraph on board Solar Orbiter [46], before the instrument was descoped by removing the spectroscopic capabilities.

### 2.1. Extraction of UV and VL Intensities

Starting from the UVCS observations, the intensity of the H VI Lyman-$\alpha$ 1215.67 Å line was measured after a standard calibration procedure by integrating over the spectral line profile, after removal of the background emission. In order to perform a combined analysis of the UVCS and Mark IV data, the pB intensities were extracted in each single exposure along the projected locations of the UVCS observations at 1.6 $R_{sun}$ and 1.9 $R_{sun}$ (white straight lines in Figure 1). Figure 2 shows the resulting evolution of the UV Lyman-$\alpha$ line (top panels) and the VL pB emission (bottom panels) at 1.6 $R_{sun}$ (left) and 1.9 $R_{sun}$ (right). All the panels in Figure 2 show the UV and VL intensity maps, with the $Y$-axis corresponding to different distances along the UVCS spectrometer slit measured from its center (see Figure 1), while the $X$-axis corresponds to different exposures/times (with time running from left to right) and has been converted into distances by assuming the speeds reported in Paper I. To facilitate the comparison between the UVCS and LASCO observations, these images have also been reflected around the slit center. Moreover, in order to increase the visibility of the CME, the pre-CME Lyman-$\alpha$ and pB-intensity latitudinal distributions have been subtracted from the following exposures.

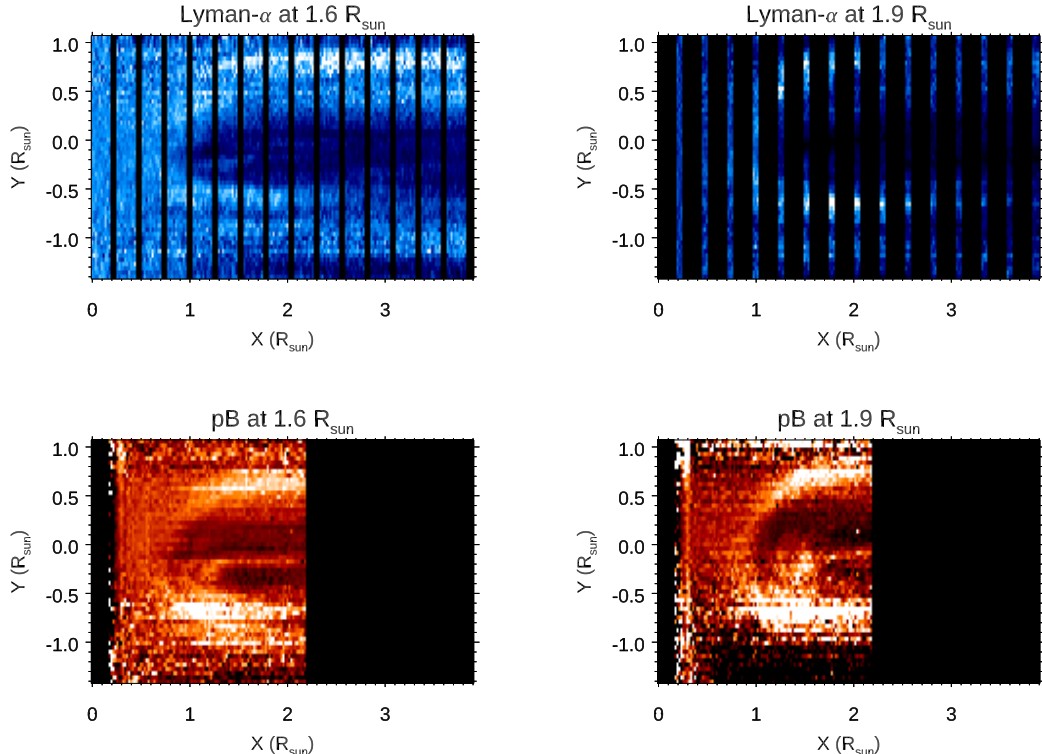

**Figure 2. Top**: the Lyman-$\alpha$ emission as observed by the UVCS instrument at 1.6 (**left**) and 1.9 (**right**) $R_{sun}$ during the transit of the CME. **Bottom**: the pB emission as observed by the MLSO Mark IV K-coronameter at 1.6 (**left**) and 1.9 (**right**) $R_{sun}$ during the transit of the CME. Pre-CME intensities have been subtracted in the Lyman-$\alpha$ and pB panels to increase the CME visibility.

As a whole, the different panels in Figure 2 clearly show the transit of the CME both in the UV Lyman-$\alpha$ and the VL pB. The CME appeared with the quite typical three-part structure, having a circular front surrounding a darker cavity and a relatively smaller and brighter core. The CME core is clearly visible in the pB images (bottom panels in Figure 2) at coordinate positions $(X, Y) = (+1.2, -0.3)$ $R_{sun}$ in the image at 1.6 $R_{sun}$ (bottom left), and $(+1.5, -0.20)$ $R_{sun}$ in the image at 1.9 $R_{sun}$ (bottom right). The CME core is also visible approximately in the same locations in the Lyman-$\alpha$ images (top panels), despite the presence of data gaps. Note that the horizontal shift along the $X$-axis by $\sim$0.3 $R_{sun}$ of

the CME core and other CME features between the left and right panels is simply due to the displacement of the CME observations from 1.6 to 1.9 $R_{sun}$.

In order to measure the 2D distribution of the plasma thermal energy density $E_{th} = 2n_e k_B T_e$ inside the CME, it is necessary to measure the 2D distributions of electron density $n_e$ and temperature $T_e$. The determination of these two quantities is described in the next two sections.

### 2.2. Electron Density Determination

The plasma electron density can be measured directly from the observed VL pB emission, which is mostly dependent on the line-of-sight (LOS) integration of $n_e$ [8], and hence on the so-called column electron density $N_e = \int n_e \, dz$. The quantity $N_e$ is usually estimated by dividing the observed pB total integrated intensity by the polarized brightness $pB_e$ of a single scattering electron [47] assumed to lie on the plane of sky (POS). This assumption is justified here because the January 31 CME appears to propagate mainly in the plane of sky (as confirmed also by the absence of Doppler shifts in the UV spectral lines—see Paper I). The derived column density $N_e$ can then be converted into electron density $n_e = N_e / L_{CME}$ by simply assuming a LOS nominal depth $L_{CME}$ of CME plasma equal to 1 $R_{sun}$ [48].

Nevertheless, to estimate the CME electron density $n_{e,CME}$ it is necessary to remove the surrounding coronal plasma not affected by the CME and aligned in each pixel along the LOS. The external coronal column densities $N_{e,cor}$ were estimated here pixel by pixel along the LOS at 1.6 and 1.9 $R_{sun}$ by assuming a typical electron density profile $n_e(r)$ from the literature [49] and by modifying the resulting column density values $N_{e,cor} = f \times \int n_e(z) \, dz$ with a constant multiplication factor $f$ until the best agreement was found for the pixels not affected by the CME. For these specific observations the best agreement (i.e., the minimum of the total differences squared) was found with a multiplication factor of $f = 0.7$. The resulting external coronal column densities $N_{e,cor}$ were thus subtracted from the observed column densities $N_{e,obs}$, and finally the CME electron densities were estimated pixel by pixel as $n_{e,CME} = (N_{e,obs} - N_{e,cor}) / L_{CME}$.

The resulting $n_{e,CME}$ maps are shown in Figure 3 (top panels). Before the arrival of the CME (left part of the maps) the coronal densities are dominated by the pre-CME coronal streamer (see also Figure 1), which is almost centered on the location of the UVCS slit center (corresponding to the origin of the $Y$-coordinates in all panels of Figure 3). Then, the arrival of the CME corresponds to a clear compression and northward deflection of the coronal streamer (positive $Y$-coordinates in all panels of Figure 3). The 2D maps clearly show again the locations of the CME front, the CME core, and intermediate CME features.

### 2.3. Electron Temperature Determination

In this study, the 2D distribution of electron temperatures $T_e$ across the CME was determined using the technique recently provided in Paper II to measure the solar wind speed, based on the so-called Doppler dimming technique [50,51]. In particular, under the assumption that the observed Lyman-$\alpha$ emission is entirely due to the radiative excitation process followed by spontaneous emission, the ratio between the observed UV Lyman-$\alpha$ ($I_{res}$) and VL pB ($I_{pB}$) intensities is almost independent of the (unknown) distribution of the electron density $n_e$ along the LOS at the altitude $\rho$, and is mostly related to the value of the neutral H ionization fraction $R_H(\rho)$ and the outflow speed $V_0(\rho)$. Hence, as discussed in Paper II, by measuring the outflow speed $V_0(\rho)$ of the observed propagating features (such as CMEs) the Doppler dimming technique can be applied to measure the plasma electron temperatures. The assumption that the observed Lyman-$\alpha$ emission is entirely due to radiative excitation is justified for the analysis of this event first by the absence of an erupting prominence associated with the CME and secondly by the low values of the ratio between the intensities of the H I Lyman-$\alpha$ and Lyman$-\beta$ lines (see discussion in Paper I). Hence, the technique recently described in Paper II can be safely applied for this CME to determine $R_H(\rho)$, according to:

$$R_H[T_e(\rho)] = \frac{H_{pB}}{H_{res}} \frac{I_{res}(\rho)}{I_{pB}(\rho)} \frac{K_{pB}(\rho)}{h(\rho)} \frac{\sqrt{\sigma_{disk}^2 + \sigma_{cor}^2(\rho)}}{\exp\left[-\frac{V_0^2(\rho)}{(\sigma_{disk}^2 + \sigma_{cor}^2(\rho))c^2/\lambda_0^2}\right]}, \tag{1}$$

where $H_{res}$, $H_{pB}$, and $K_{pB}$ terms include all the different physical constants and quantities required for the determination of the UV Lyman-$\alpha$ ($I_{res}$) and VL pB ($I_{pB}$) intensities, $h(\rho)$ is a geometrical function related to the solid angle subtended by the solar disk at the scattering point, and $\sigma_{disk}$ and $\sigma_{cor}$ correspond to the $1/e$ half-widths of the chromospheric and coronal Lyman-$\alpha$ profiles (see Paper II for details). In particular, in this study, the proton kinetic temperature $T_p$ (related to $\sigma_{cor}$) was assumed to be equal to the electron temperature $T_e$ in a coronal streamer at the projected observation altitudes along the UVCS slit, as given in [52], while the Lyman-$\alpha$ disk intensity $I_0$ (related to the $H_{res}$ term) was assumed to be in agreement with Paper I. A constant outflow speed $V_0$ was assumed at each altitude, equal to an average value of the measured CME front and core speed. From the values of $R_H$ determined with the above equation, the electron temperatures $T_e$ can be measured by considering that for $T_e$ between $10^6$ and $10^8$K, the neutral H ionization equilibrium curve provided by the CHIANTI spectral code [53] can be fitted to about 10% accuracy using

$$T_e \simeq 0.59\, R_H^{-0.9407}, \tag{2}$$

as given by [54]. The resulting 2D electron temperature maps are shown in Figure 3 (bottom panels) and discussed in the next section.

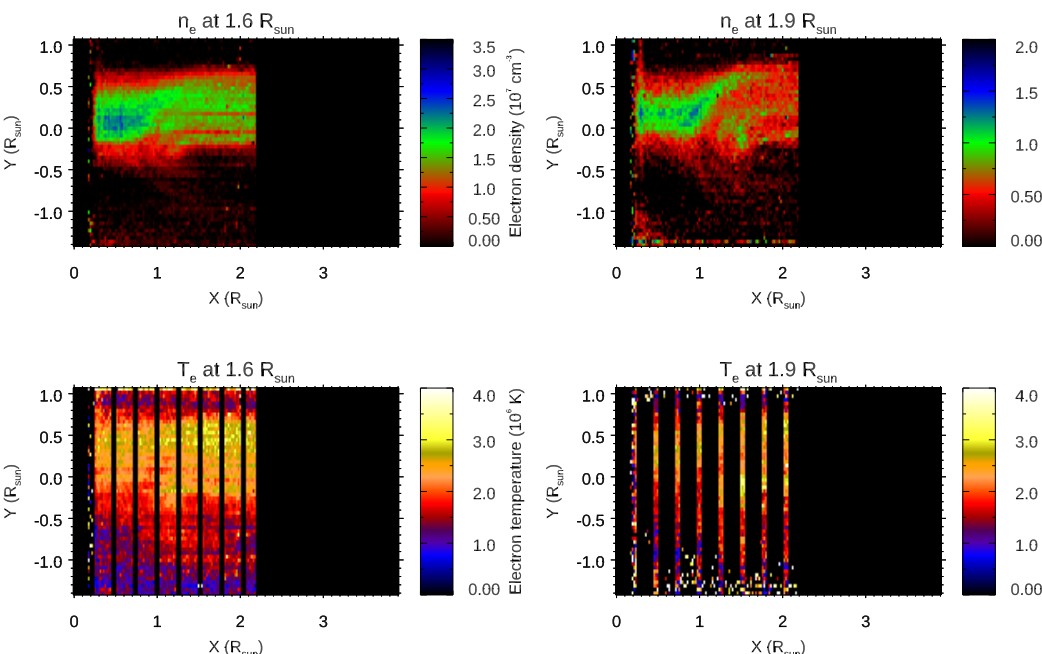

**Figure 3.** Maps showing the 2D distributions of electron densities (**top**) and electron temperatures (**bottom**) as derived from the analysis of data acquired during the CME transit at 1.6 $R_{sun}$ (**left**) and 1.9 $R_{sun}$ (**right**).

## 3. Results

Before discussing the results presented here, it is very important to clarify the differences between the method applied for the temperature determination in Paper I and the new method described in Paper II and applied in this study. In both cases, the plasma temperatures were determined from a comparison between the UV Lyman-$\alpha$ and VL pB intensities. However, in Paper I, the electron densities were first derived from the VL pB observations, and then these densities were employed to estimate the expected value of the UV Lyman-$\alpha$ modifying the plasma temperatures, until the best agreement between the cal-

culated and the observed intensities was found. The problem with this technique is that the temperature measurement depends on the previous density measurement, and a possible overestimate (underestimate) of the density corresponds to a overestimate (underestimate) also of the temperature. On the other hand, in this study, the electron temperatures were derived using the direct ratio technique described in Paper II. This has the advantage that the temperature determination is not related to possible uncertainties in the density determination. Moreover, the method described in Paper II is more suitable for the construction of 2D temperature maps (as needed to study CME thermodynamic evolution), because the temperatures are determined from the direct ratios between 2D intensity maps.

To facilitate the discussion, the results given by the 2D maps in Figure 3 are also shown more quantitatively as 1D plots in Figure 4. In particular, the density and temperature profiles in Figure 3 were extracted at constant latitudes corresponding to the transit of the CME core at 1.6 and 1.9 $R_{sun}$. Note that, unlike the original report in Paper I, the transit of the CME core at 1.9 $R_{sun}$ was not missed due to a data gap but instead was also observed at this altitude in a very few exposures acquired by UVCS. This allowed the study of the variation with altitude of plasma temperatures inside different parts of the CME and, in particular, in the CME core.

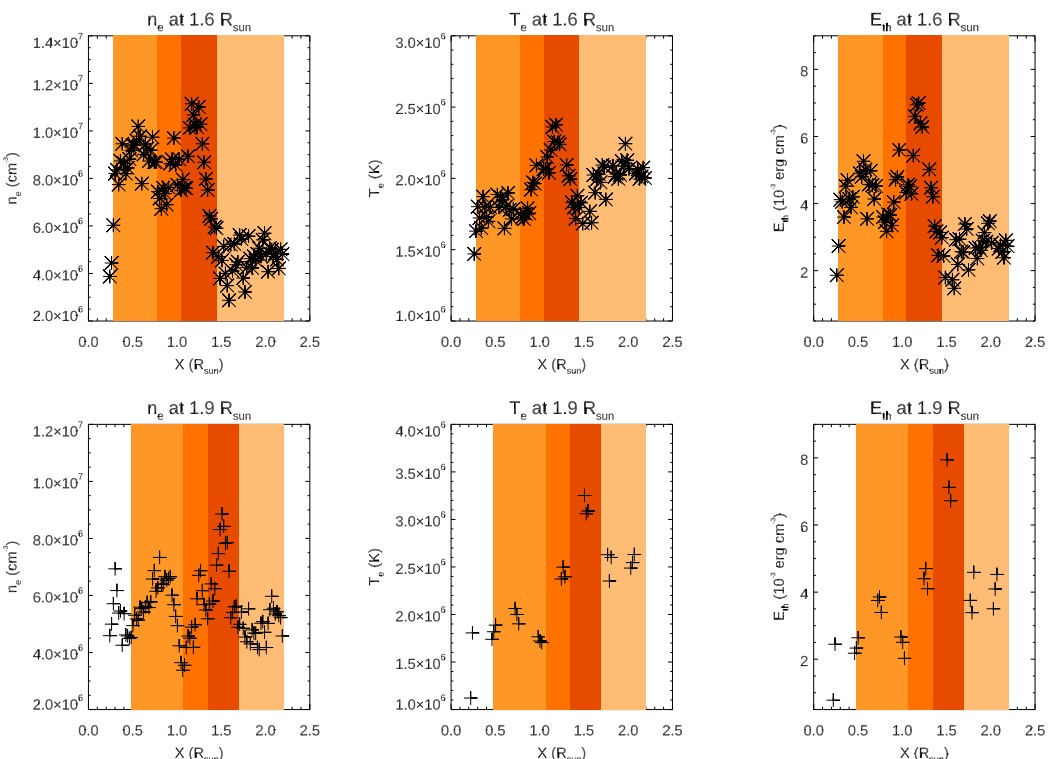

**Figure 4.** Plots showing the 1D distributions of electron densities (**left** column), electron temperatures (**middle** column), and thermal energy densities (**right** column) at 1.6 (**top** row) and 1.9 (**bottom** row) $R_{sun}$, extracted at the latitude where the CME core transit was observed. Different colors indicate (from **left** to **right**) the identified locations of the CME front, void, core, and post-CME plasma.

The 1D plots in Figure 4 (left panels) show, with different background colors, the corresponding locations of these different CME parts, i.e., the CME front, void, core, and post-CME corona, as identified from the evolution of the plasma densities at both 1.6 and 1.9 $R_{sun}$. These density curves were obtained from the MLSO Mark IV data, with continuous data coverage, and hence the density maps (Figure 3) and curves (Figure 4) have no data gaps and are suitable for the identification of CME parts. On the other hand, the UVCS observations were acquired, as previously mentioned, alternately at 1.6 and 1.9 $R_{sun}$, and the resulting temperature maps (Figure 3) and curves (Figure 4) have data gaps. Nevertheless, the very interesting result of this study is obtained once the same locations

for the different CME parts as determined from the density curves (left panel in Figure 4) are superposed on the temperature curves (middle panel in Figure 4).

In this study, it is assumed that the main sources of uncertainties are related to the radiometric calibrations of different instruments. In particular, the UVCS radiometric calibration uncertainty is about 22% for the first-order lines [55], while the radiometric calibration uncertainty for the MLSO Mark IV K-coronameter is about 15% [3]. By propagating these uncertainties in the above equations for density and temperature measurements, the resulting uncertainty in the measured temperatures becomes about 36%, while a value of 15% can be assumed for the uncertainties in the measured densities. Note that the Lyman-$\alpha$ and pB intensity variations shown in Figure 2 are not larger than 20–30%, and hence are comparable with the above uncertainties. This reminds us that these uncertainties are associated with systematic and not random errors in the measurements. All the other assumed quantities also lead to possible smaller but systematic errors.

This superposition shows first of all that (as already reported in Paper I) the front temperatures are larger at both altitudes than the pre-CME coronal temperatures. Considering the density ratio between the CME front and the pre-CME corona, the temperature increases were confirmed here to be compatible with a simple adiabatic compression, as also reported by Paper I. In particular, at 1.6 $R_{sun}$, starting from a pre-CME coronal density of $n_{cor} \simeq 7 \times 10^6$ cm$^{-3}$ and a coronal temperature of $T_{cor} \simeq 1.5 \times 10^6$ K (in very good agreement, for instance, with measurements by [49]), and considering a CME front density of $n_{front} \simeq 1 \times 10^7$ cm$^{-3}$, the observed CME front temperature $T_{front} \simeq 1.9 \times 10^6$ K is in agreement with the expected temperature increase due to adiabatic compression $T_{front} = T_{cor}(n_{front}/n_{cor})^{2/3}$ (having assumed $\gamma = 5/3$). This is also confirmed at 1.9 $R_{sun}$, with $n_{cor} \simeq 5 \times 10^6$ cm$^{-3}$, $T_{cor} \simeq 1.7 \times 10^6$ K, $n_{front} \simeq 7 \times 10^6$ cm$^{-3}$, and $T_{front} \simeq 2.1 \times 10^6$ K (see Figure 4). Hence, the plasma heating in the CME front is compatible at both altitudes with adiabatic compression alone.

A second interesting result shown in Figure 4 is that, in agreement again with Paper I, the temperatures increase continuously inside the CME bubble from the CME front to the CME core. At both altitudes, the peak temperature is reached in the CME core, with $T_{core} \simeq 2.4 \times 10^6$ K at 1.6 $R_{sun}$ and $T_{core} \simeq 3.2 \times 10^6$ K at 1.9 $R_{sun}$. These measurements also provide a new result that was not provided in Paper I: the CME core temperature increases during the CME expansion, providing a clear signature of plasma heating processes occurring. It is very important to notice here that the CME core temperature increases despite the plasma density decrease, and hence despite the expected adiabatic expansion cooling (which is likely to be the dominant physical process responsible for thermal energy variations). In particular, starting from the above core temperature at 1.6 $R_{sun}$ and a density $n_{core} \simeq 1.1 \times 10^7$ cm$^{-3}$, and considering that at 1.9 $R_{sun}$ the core density has decreased to $n_{core} \simeq 9 \times 10^6$ cm$^{-3}$, by assuming adiabatic cooling the expected core temperature at this altitude should be $T_{core} \simeq 2.1 \times 10^6$ K, instead of the observed value of $T_{core} \simeq 3.2 \times 10^6$ K.

It is also interesting to consider the evolution of the thermal energy density $E_{th} = 2 n_e k_B T_e$ inside the CME; the corresponding curves for this quantity are given in Figure 4 (right panels). Focusing on the CME core, because the core density decreases by a factor ~1.2, while the core temperature increases by a factor ~1.3, it is found that the thermal energy density in the CME core increases by about ~10% between 1.6 and 1.9 $R_{sun}$. In particular, the thermal energy density increases between the two observed heliocentric distances from $E_{th} = 7 \times 10^{-3}$ erg cm$^{-3}$ (or $5.7 \times 10^{14}$ erg g$^{-1}$) at 1.6 $R_{sun}$, up to $E_{th} = 8 \times 10^{-3}$ erg cm$^{-3}$ (or $7.9 \times 10^{14}$ erg g$^{-1}$) at 1.9 $R_{sun}$. These values are comparable, for instance, with the total cumulative heating found in [28] for another event and with thermal energies for some of the blobs studied in [30]. This thermal energy density increase is opposite to that observed in the CME front, where the thermal energy density instead decreases, and it clearly indicates that inside the CME core there are physical processes other than adiabatic compression/expansion playing a role in the observed plasma heating. If this thermal energy density increase is directly ascribed to the dissipation of magnetic energy density $E_m = B^2/2\mu$, a variation of $\Delta E_{th} = 10^{-3}$ erg cm$^{-3}$ corresponds to the dissipation of an internal magnetic field of the order

of $B_{core} = \sqrt{8\pi\,\Delta E_{th}} \simeq 0.16$ G. Since the CME core propagated from 1.6 to 1.9 $R_{sun}$ in about $\Delta t = 2470$ s, the resulting heating rate is of the order of $\Delta E_{th}/\Delta t \sim 4 \times 10^{-7}$ erg cm$^{-3}$s$^{-1}$.

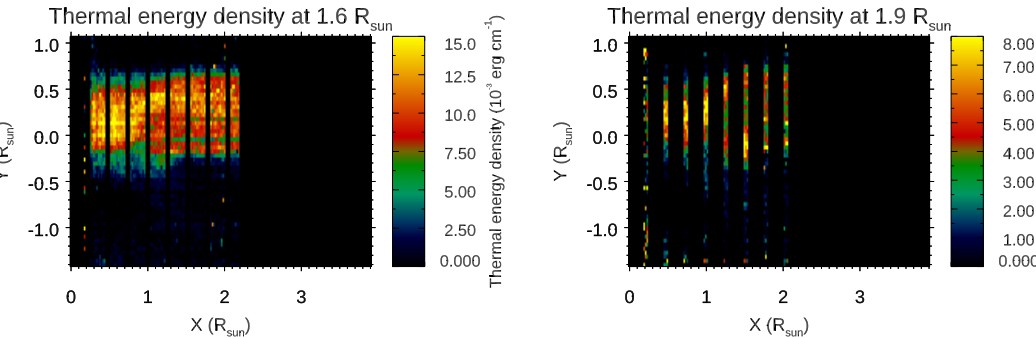

**Figure 5.** Resulting 2D distributions of the plasma thermal energy density during the CME transit as observed at 1.6 (**left**) and 1.9 (**right**) $R_{sun}$.

The corresponding 2D evolutions of the thermal energy densities at the two altitudes are also provided in Figure 5. The 2D images also allow the estimation of the possible expansion rate of the CME core volume: by considering (from Figures 4 and 5) the apparent 2D project core elliptical diameters $(d_X, d_Y)$ approximately equal to $(0.4, 0.2)$ $R_{sun}$ at 1.6 and $(0.35, 0.4)$ $R_{sun}$, and by assuming $d_Z = d_X$, it is possible to estimate the CME core volume as $V_{core} = \pi/6\,d_Y d_X^2$, leading to $V_{core} = 5.7 \times 10^{30}$ cm$^3$ and $V_{core} = 8.7 \times 10^{30}$ cm$^3$ at 1.6 and 1.9 $R_{sun}$ respectively. With these values, by assuming the above peak thermal energy densities to be uniformly distributed in the CME core (probably providing just an upper-limit estimate), the total resulting thermal energy content $U_{th}$ is found to be $U_{th} = 4 \times 10^{28}$ erg and $U_{th} = 7 \times 10^{28}$ erg at 1.6 and 1.9 $R_{sun}$, respectively. Hence, the total extra energy deposited in the CME core volume is of the order of $3 \times 10^{28}$ erg.

These maps also show that at both altitudes/times the 2D distribution of thermal energy has a low level of symmetry with respect to the CME propagation axis. This is due to the fact that both the electron temperatures and densities have an imbalance in the void region surrounding the CME core, appearing to be higher in the northward half of the CME (Figure 6). This could be partly related with projections effects, if the surrounding coronal streamer being crossed by the CME also contribute along the LOS to the observed UV and VL emissions. Alternatively, this could also be a dynamic effect related with the physical interaction of the CME flux-rope expanding against the nearby coronal streamer. The flux-rope should be magnetically isolated with respect to the surrounding corona, but the observed density and temperature imbalance could be related with a pressure imbalance related with the CME-streamer interaction.

## 4. Summary and Conclusions

This work demonstrates again the potential of UV and VL data combination to determine the thermodynamic evolution of coronal mass ejections (CMEs). Using unique observations acquired on 31 January 2000 by the UVCS spectrometers of the same CME at two different heliocentric distances (1.6 and 1.9 $R_{sun}$), it was possible to investigate (in combination with MLSO Mark IV K-coronameter observations) the thermal evolution of the same CME structures, particularly the CME front, void, and core. The results at both altitudes showed that the temperature increases observed in the front with respect to the pre-CME plasma were compatible with a simple adiabatic compression. This is in agreement with the usual idea that the CME front is formed mainly by piling up of the overlying coronal plasma met by the expanding CME flux rope [56].

Then, the plasma temperatures progressively increase further, moving inside the CME bubble and crossing the void, finally reaching a maximum at the CME core. Due to the availability of the UVCS observations of the same CME core at two different altitudes, it was possible to follow the temperature and thermal energy evolution. The results showed

that the CME core temperatures increased during the expansion, despite the expected plasma cooling, mainly due to adiabatic expansion. The derived variation of the thermal energy density allowed the estimation that, under the hypothesis that this energy originates from the dissipation of magnetic energy (with conservation of magnetic helicity) according to the model provided in [33], the strength of the dissipated magnetic field inside the CME core should be of the order of ∼0.16 G. The inferred additional plasma heating inside the CME core is comparable with or even higher than previous observations reported in [28,30].

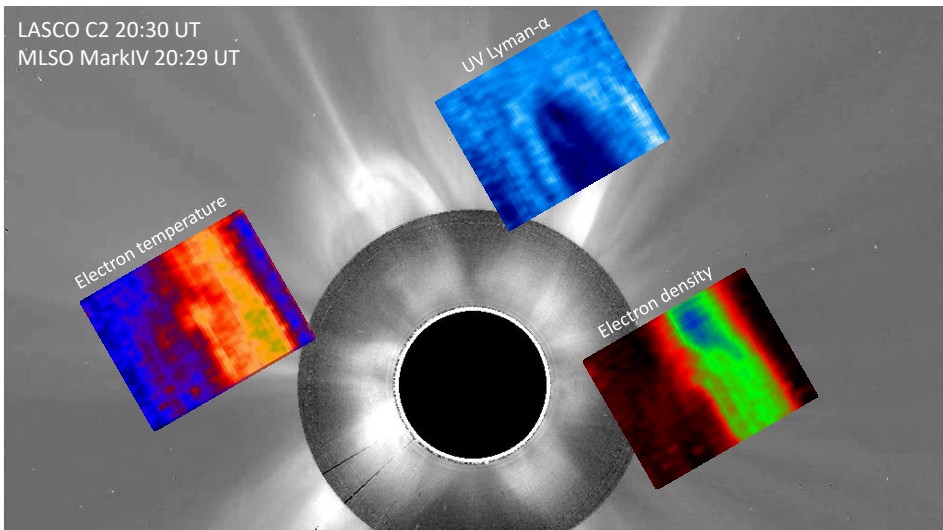

**Figure 6.** Combined image showing a comparison between the CME as observed from MLSO Mark IV K-coronameter and SOHO/LASCO-C2 coronagraph (background image) and the corresponding distribution of UV Lyman-*α* intensity (top right), electron densities (bottom right), and electron temperatures (bottom left). For this comparison the data gaps have been eliminated and the images have been rotated clockwise by 60° with respect to those shown in Figures 2 and 3.

Overall, the results presented here show the potential of present and future observations provided by multi-channel coronagraphs observing at the same time the UV Lyman-*α* and the VL pB emissions. This is summarized in Figure 6, providing a one-to-one correspondence between the CME observed in the UV Lyman-*α* by UVCS and in the VL pB observed by SOHO LASCO and MLSO Mark IV instruments, as well as the resulting distributions of plasma densities and temperatures across the CME and the nearby coronal streamer. The results presented here provide the first 2D maps of temperatures and thermal energy densities inside a CME observed in the intermediate corona. Such temperature and density measurements are usually performed with full-disk EUV imagers, but are limited to the inner corona (below 1.5 $R_{sun}$) [21,22]. Higher up (above 1.5 $R_{sun}$), CME temperatures have been measured using the UVCS spectrometer, but the analysis is usually limited to a single CME region or specific times during the observations (see the review in [27], Section 6.1.3). Moreover, these 2D maps are provided here for the first time at two different altitudes, corresponding to two different times in the CME evolution, allowing the extension of the results provided in Paper I regarding the temporal evolution of CME thermodynamic properties.

Similar analyses will be applied also to the observations of CMEs currently being acquired [57] by the Metis instrument [58] on board Solar Orbiter, when the radiometrically calibrated UV observations will be officially distributed, as well as to the observations that will soon be acquired by the Lyman-*α* Solar Telescope (LST) on board the forthcoming Chinese ASO-S mission coronagraph [59]. Moreover, to the best of the author's knowledge, the results presented here represent the only existing "multi-slit-like" study of a CME, and show the importance of future instrumentation having the capability to perform multi-slit spectroscopic observations of CMEs and other coronal phenomena, as originally proposed in [46].

**Author Contributions:** The research described here was entirely carried out by the sole author of this article. All authors have read and agreed to the published version of the manuscript.

**Funding:** This research received no external funding.

**Institutional Review Board Statement:** Not applicable.

**Informed Consent Statement:** Not applicable.

**Data Availability Statement:** The SOHO UVCS and LASCO data can be freely downloaded from VSO archive (December 2021) https://sdac.virtualsolar.org/cgi/search (accessed on 1 December 2021). The MLSO Mark IV data can be freely downloaded from the MLSO website (December 2021) https://mlso.hao.ucar.edu/mlso_data_calendar.php?calyear=2000&calinst=mk4 (accessed on 1 December 2021).

**Acknowledgments:** This work was originally initiated during a visit of the author to the Harvard-Smithsonian Center for Astrophysics, whose hospitality and support are gratefully acknowledged. The author would also like to thank G. Poletto and J.C. Raymond for their collaboration and help on the determination of previous results of this work.

**Conflicts of Interest:** The author declares no conflict of interest.

## Abbreviations

The following abbreviations are used in this manuscript:

| | |
|---|---|
| CME | Coronal mass ejection |
| DEM | Differential emission measure |
| EUV | Extreme ultraviolet |
| FOV | Field of view |
| ICME | Interplanetary coronal mass ejection |
| LOS | Line of sight |
| MLSO | Mauna Loa Solar Observatory |
| POS | Plane of the sky |
| UV | Ultraviolet |
| UT | Universal Time |
| VL | Visible light |
| VSO | Virtual Solar Observatory |

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
