# Peer review of "Temperature and Thermal Energy of a Coronal Mass Ejection"

_symmetry, doi:10.3390/sym14030468_

Round 1
Reviewer 1 Report
The article "Temperature and thermal energy of a Coronal Mass Ejection" is an interesting study about estimating the temperature profile of CME's. The manuscript is well written and organized. I have only few minor comments (listed below). After these slight clarifications, the manuscript could be accepted.
- Some figure are (mainly Fig. 1 and 2) small. Is it possible to increase their sizes?
- Page 5, line 169. You mentioned that you used a typical electron density profile. Could you open this? It is not fully clear and accepted view, which is a typical model.
- Could you define some error limits (values) for your temperature (for example page 8, line 240) etc. estimations?
- Please check that "Rsun" is written consistently over the whole manuscript.
Author Response
The article "Temperature and thermal energy of a Coronal Mass Ejection" is an interesting study about estimating the temperature profile of CME's. The manuscript is well written and organized. I have only few minor comments (listed below). After these slight clarifications, the manuscript could be accepted.
1. Some figure are (mainly Fig. 1 and 2) small. Is it possible to increase their sizes?
> Probably the Referee refers to Figs. 2 and 3, not 1 and 2. According to this comment, all the panels in these Figures have been rotated by 90 degrees and the formats of the Figures have been changed. Color bars have been added as well. This rotation also facilitate a direct comparison with plots given in Fig. 4. All the corresponding changes in the text referring to these Figures have been applied.
2. Page 5, line 169. You mentioned that you used a typical electron density profile. Could you open this? It is not fully clear and accepted view, which is a typical model.
> Maybe the Referee missed the provided reference to paper [49] corresponding to Gibson et al. (1999).
3. Could you define some error limits (values) for your temperature (for example page 8, line 240) etc. estimations?
> A full paragraph discussing possible uncertainties has been added.
4. Please check that "Rsun" is written consistently over the whole manuscript.
> Done thanks, there was only one mistake in the abstract.
Reviewer 2 Report
I read with pleasure the manuscript "Temperature and thermal energy of a Coronal Mass Ejection" and I suggest it is accepted after some minor points have been addressed.
- Line 54: In particular, the analysis ….. demonstrated .. (remove the second comma and add past tense ending)
- Line 59: …but theoretical considerations [32]…. (Remove “the”)
- The final part of the introduction would benefit from some explanation of what is new in this paper with respect to papers I and II (which you basically do at the beginning of Sec 3, Results)
- Line 90: These kinds….
- Line 113: can you explain a bit more why this is the case, I.e. why in the outer part of the domain the errors are higher? Also: this means that you are not considering almost half of the domain inspected by the observations, between 2 and 2.79 R_sun. How does this affect the results?
- Line 171: You mention “best agreement”. Which kind of fit did you perform? Is the 0.7 factor consistent with previous studies of CME column densities?
- In eq. 1 what geometrical function h(ρ) did you use?
- In Sec. Summary, whilst the combined use of UV and VL data is very well stressed, it is not clear what advances this work produces with respect to previous studies, such as papers I, II, and other papers cited (e.g. refs 28, 30). Could you add a couple of sentences on this matter?
- Can your analysis say something about the helicity content in CME? And about the magnetic energy dissipated via magnetic reconnection? You could perhaps add something about these very important features in CME.
Author Response
I read with pleasure the manuscript "Temperature and thermal energy of a Coronal Mass Ejection" and I suggest it is accepted after some minor points have been addressed.
- Line 54: In particular, the analysis ….. demonstrated .. (remove the second comma and add past tense ending)
> Done thanks.
- Line 59: …but theoretical considerations [32]…. (Remove “the”)
> Done thanks.
- The final part of the introduction would benefit from some explanation of what is new in this paper with respect to papers I and II (which you basically do at the beginning of Sec 3, Results)
> A sentence has been added to better outline the novelty of the analysis presented here.
- Line 90: These kinds….
> Done thanks.
- Line 113: can you explain a bit more why this is the case, I.e. why in the outer part of the domain the errors are higher? Also: this means that you are not considering almost half of the domain inspected by the observations, between 2 and 2.79 R_sun. How does this affect the results?
> This is a standard limit for ground-based coronagraphs such as Mark IV: due to the sky brightness it's very hard to get coronagraphic observations at larger altitudes (see also the left panel of Figure 10 in the Bemporad et al. 2007 paper) because the sky becomes too bright with respect to the corona. In any case the results presented here are not affected at all by this problem, in fact (as explained in the text) the pB intensities have been extracted along the projected locations of the UVCS observations (white lines in Fig. 1) and these are almost always located below 2 Rsun, with the only exception of a few pixels at the southward edge of the slit located at 1.9 Rsun (see Fig. 1), where in any case the CME transit was not observed.
- Line 171: You mention “best agreement”. Which kind of fit did you perform? Is the 0.7 factor consistent with previous studies of CME column densities?
> A small sentence was added in brackets to better explain this. This factor - as explained - is not needed to reproduce the CME column density, but to reproduce the column density of the surrounding background corona to be removed in order to isolate the CME excess column density.
- In eq. 1 what geometrical function h(?) did you use?
> A small sentence has been added to better explain what this geometrical function is. In order to keep shorter the description of the analysis presented here, I refer the Referee and reader to Paper II for the explicit expressions of these therms (to help the comparison I used exactly the same therminology in the two papers).
- In Sec. Summary, whilst the combined use of UV and VL data is very well stressed, it is not clear what advances this work produces with respect to previous studies, such as papers I, II, and other papers cited (e.g. refs 28, 30). Could you add a couple of sentences on this matter?
> Results presented here provide the first ever made 2D maps of temperatures and thermal energy densities inside a CME expanding in the intermediate corona; moreover, these maps are provided for the first time at two different altitudes corresponding to two different times in the CME evolution. Such measurements are usually performed only in the inner corona (1.3 Rsun) with full-disk EUV imagers, or higher up with the UVCS spectrometer (>1.5) but the analysis is usually limited to single CME regions or specific times during the observations. In short, a Figure like Figure 6 was never provided in the literature before. These points are now better stressed in the Summary.
- Can your analysis say something about the helicity content in CME? And about the magnetic energy dissipated via magnetic reconnection? You could perhaps add something about these very important features in CME.
> An estimate of the dissipated magnetic field is already provided in Section 3 and the value of 0.16 Gauss was obtained. Unfortunately, to the knowledge of the author there are no existing measurements of magnetic fields inside expanding CMEs in the intermediate corona to be compared with this estimate. No information can be derived about the magnetic helicity, the only this that I added in the paper is that in the Kumar & Rust (1996) model the magnetic energy is dissipated, but the magnetic helicity is conserved.